# Repurposing of Anti-Malarial Drug Quinacrine for Cancer Treatment: A Review

Makhan Kumar and Angshuman Sarkar *

CMBL, Department of Biological Sciences, Birla Institute of Technology and Sciences, K K Birla Goa Campus, NH17B, Zuarinagar, Sancoale 403726, Goa, India; p20150011@goa.bits-pilani.ac.in
* Correspondence: asarkar@goa.bits-pilani.ac.in

**Abstract:** Quinacrine (QC), a synthetic drug belonging to the 9-aminoacridine family, has been used extensively to treat malaria and multiple ailments over the past several decades. Following its discovery in the 1920s and extensive use for the treatment of malaria for nearly two decades, numerous studies have explored its antineoplastic potential in both preclinical and clinical settings. Multiple studies spanning over seven decades have examined a wide range of QC anticancer activities across various types of cancers, along with the underlying mechanisms. Many of these mechanisms, including activation of the p53 signaling cascade and simultaneous NF-κB signaling inhibition, have been reported in various studies, bringing QC to a unique polypharmacological category drug possessing the potential to treat a wide variety of diseases, including cancer. This article summarizes most of the research conducted over several decades to uncover new molecular mechanisms activated or inactivated and directly correlate with antineoplastic activity QC.

**Keywords:** quinacrine; cancer treatment; apoptosis; cell cycle arrest; autophagy

## 1. Introduction

Quinacrine (IUPAC name: 4-N-(6-chloro-2-methoxyacridin-9-yl)-1-N,1-N-diethylpentane-1,4-diamine) is a synthetic drug belonging to the family of acridine-based synthetic compounds and the 9-aminoacridine subfamily. The chemical structure of QC is composed of three heterocyclics with acridine, which was designed and produced by the pharmaceutical organization Bayer in 1928 during the post-World War I period and was actively used for four years during World War II to treat soldiers infected with malaria [1]. According to a U.S. Surgeon General Office report, more than three million soldiers had reportedly taken QC during this period, and they were routinely monitored for safety and efficacy, making QC one of the most extensively studied synthetic drugs in the history of modern medicine [2,3]. Low concentrations of QC have been shown to induce redistribution of $Ca2^+$ ions, leading to disruption of IP3 dependent $Ca2^+$ oscillation, hampering the growth of *Plasmodium falciparum*, as they exhibit stage-specific $Ca2^+$ oscillations in the ring form during the trophozoite stage, which is a key requirement for the maturation of the parasite inside erythrocytes. Disruption of these oscillations obstructs intra-erythrocytic maturation, resulting in the death of *P. falciparum* parasites [4]. Chloroquine eventually replaced QC, which has a similar structure, but a quinolone replaced the central ring. Since the World War II era, QC has been explored and utilized for treating various infections and other ailments, including giardiasis, taeniasis, helminthic infections, prion diseases, and lupus erythematosus, and as anti-inflammatory drugs for rheumatoid arthritis [5–8]. QC has shown significant clinical efficacy in treating all these ailments, especially lupus and rheumatoid disease. Exploring QC activity and usage beyond treating the conditions and infections mentioned earlier started when its DNA-binding and fluorescent properties were discovered in the early 1950s. Owing to its fluorescent and DNA-binding properties, QC is also routinely used in laboratories to stain chromosomes and study their patterns. This

technique is explicitly called "Q-banding" [9–11]. In addition, QC was extensively utilized as a sterilizing agent in the 1980s [12,13].

## 2. Methods

A systemic search with the title "Quinacrine and cancer research" was conducted in PubMed and the relevant articles were saved onto the drive, thoroughly analyzed for quality and key findings. The saved articles were indexed and stored for future study.

## 3. Internalization and Pharmacokinetics of QC

Quinacrine is prescribed as oral tablets to be taken with water after a meal; however, other administration routes, such as intravenous, intramuscular, rectal, transcervical, and interstitial, have also been used to deliver the drug [8,14,15]. QC is rapidly absorbed following oral administration, and its serum plasma concentration peaks approximately 12 h post-ingestion [3]. The plasma concentration of QC increases during the first week of administration and equilibrates at 94% around the fourth week of treatment. During standard treatment cycles, peak plasma concentrations of up to 140 ng/mL (0.32 $\mu$M) have been reported during its utilization to treat malaria [16]. Nearly 80–90% of the drug present in the plasma is bound to plasma proteins, and the half-life of the drug inside the body is roughly around fourteen days. QC is evenly distributed throughout the body, with the liver, kidney, lungs, and spleen having the highest concentrations. In contrast, the heart, brain, and skeletal muscle show the lowest QC concentrations. The primary route of eliminating QC is through the renal system, which can be altered depending on clinical requirements, that is, enhanced through acidification or reduced through alkalinization [17].

Quinacrine is mainly internalized through P-glycoprotein ABC transport pumps and vacuolar ATPases (Vav-ATPases) [18]. Continuous pumping by Vav-ATPases maintains the drug in a non-diffusible and cationic form in large vacuoles. Concentrations as low as 25 nM have been shown to be readily taken up by cells from 30 min to 3 h. The maximum uptake of QC varies depending on cell type. The smooth muscle cells showed maximum uptake at 5 $\mu$M concentration over two hours. The concentration-dependent 30 min uptake of QC follows hyperbolic kinetics, with a $K_m$ of 8.7 $\mu$M in umbilical smooth muscle cells, 1.14 $\mu$M in peripheral mono nucleated leukocytes (PMLNs), and 6.32 $\mu$M in lymphocytes [19]. Upon internalization, QC is readily metabolized to monomethyl quinacrine by cytochrome P450 isoform CYP3A4/5 [20]. A study of the tissue distribution of QC analogs and hydroxychloroquine showed that QC exhibited a higher concentration in A549 induced tumors than HCQ and its analog VATG-027. The mean tumor concentration for QC after the 28-day treatment was found to be approximately 1200 ng/mL and reached statistical significance. They also demonstrated that QC drug retention in lung tissue was significantly higher (2600 ng/mL $\pm$ 580 ng/mL), only behind that in the liver and kidney tissues [21].

*Reported Toxicities of Quinacrine*

QC is one of the most extensively used drugs and possesses certain advantages in terms of drug tolerance and toxicity in a great number of patients, unlike any other medicine. QC is generally well tolerated and considered safe at clinical dosages of 400 mg daily. However, a few notable toxicities have been observed in a smaller fraction of patients and those treated with high-dose regimens. Dermatological toxicity is one of the most prominent and visible side effects in patients on a standard drug regimen as it accumulates in the skin, producing yellowish-appearing stains and blue and black rashes surrounding the stained areas, presumably due to its melanin-binding properties. However, the generation of rashes has been found in a significantly lower percentage of the population (1.6%), based on a study conducted on 120,000 Australian soldiers during World War II [22]. The most threatening toxic side effect of QC is anaplastic anemia, which occurs in a significantly lower percentage (0.003%) of patients at higher dosages than recommended and with less frequent blood counts during long periods of treatment [23–25]. However, in modern

clinical studies on QC, a dose of 300 mg has been found to be well tolerated, and no incidences of anaplastic anemia have been reported. In rare cases, patients administered a dose of over 500 mg might experience drug-induced hypersensitivity in the cornea, which is primarily reversible [26,27]. Toxicities in the central nervous system have also been reported in rare cases [28,29]. However, they occurred in sporadic cases (0.1–0.4%), as only twenty-eight out of nearly 300,000 soldiers treated with QC experienced psychotic effects [30]. Apart from these side effects, some general side effects, including mild headache and gastrointestinal problems such as diarrhea, nausea and abdominal cramps, can be experienced and are primarily minor and reversible upon dose reduction.

## 4. Quinacrine and Cancer

The first study that tested the efficacy of QC in cancer was published in 1958 [31], and Vassey et al. reported a significant increase in the survival of carcinoma tumor-bearing C57BL mice following QC administration. Another study conducted by Hill and colleagues tested QC for breast cancer in laboratory and animal models. It demonstrated a similar growth inhibition pattern in cancer cells in both in vitro and in vivo models [32]. A few clinical studies on QC for the treatment of malignant pleural effusions were conducted on smaller subsets of patients (<100) over the next two decades [14,33–35]. QC was reported to be a better alternative to bleomycin in one of the studies owing to its better response and fewer side effects displayed by the patients enrolled in the study. These studies paved the way for exploring the antineoplastic potential of this antimalarial drug. Interest in understanding the molecular mechanisms involved in the antineoplastic effect of QC rose in the late 1990s. Since then, many new QC mechanisms have been reported over a wide range of cancers, demonstrating its multi-spectrum effectivity and propensity to modulate the expression of multiple signaling molecules that regulate one or more oncogenic signaling networks. Numerous preclinical studies have demonstrated a significant reduction in cancer cell viability owing to QC exposure in both in vitro and in vivo models across the spectrum of various cancer types. The reported $LD_{50}$ values have been shown to vary from t2 to 10 µM depending on the type of cancer cells [36,37]. Many new mechanisms of QC activity that have a direct impact on cancer cell proliferation and spread have been uncovered (Table 1). These studies have highlighted the activity and role of QC in regulating various cellular phenomena such as DNA replication, epigenetic regulation including reversal of gene expression of methylated or silenced genes, cell cycle progression, and interference with signaling pathways leading to cell proliferation. Accumulation of these molecular effects contributes to the pharmacological activity of QC. The effects of QC on these processes are described in detail in the following subsection.

**Table 1.** Table summarizing the effect of QC studied on various types of cancers. NSCLC—Non-small Cell Lung Cancer, CC—Colon Cancer, ATC—Anaplastic Thyroid Cancer, GBM—Glioblastoma Multiforme, BC—Breast Cancer, AML—Acute Myeloid Leukemia, HCC—Hepatocellular Carcinoma, OC—Ovarian Cancer, RCC—Renal Cell Carcinoma, HNSCC—Head and Neck Squamous Cell Carcinoma, GIST—Gastrointestinal Tumor, ALL—Acute Lymphocytic Leukemia, BER—Base Excision Repair.

| Sl No. | Type of Cancer | Drugs Studied | Effects | Reference |
|---|---|---|---|---|
| 1 | NSCLC | Quinacrine | Inhibits GSTA1 activity, ROS generation, and mitochondrial damage | [38] |
| 2 | NSCLC | Quinacrine | Suppresses NF-κB activity and ICAM-1 transcription | [39] |
| 3 | NSCLC | Quinacrine and Erlotinib | Inhibits FACT and NF-κB activity and cell cycle arrest and restores sensitivity to Erlotinib | [40] |
| 4 | CC | Quinacrine and TRAIL | Downregulation of c-FLIP and Mcl-1, sensitizes cells to TRAIL induced apoptosis | [41] |

**Table 1.** *Cont.*

| Sl No. | Type of Cancer | Drugs Studied | Effects | Reference |
|---|---|---|---|---|
| 5 | CC | Quinacrine | Upregulates p53 and p21 expression, simultaneous downregulation of p62 SQSTM leading to autophagic cell death | [42] |
| 6 | CC | Quinacrine | Downregulates p-Chk1/2, increases binding activity between p-Chk1/2 and β-TrCP leading to its degradation, $G_2/M$ arrest, apoptosis | [43] |
| 7 | ATC | Quinacrine and Sorafenib | Eradicates Mcl-1 expression, arrests proliferation and apoptosis induction | [44] |
| 8 | GBM | Quinacrine and TRAIL | Enhancement of TRAIL sensitivity and apoptosis induction | [45] |
| 9 | GBM | Quinacrine and telomozideQuinacrine and SI113 | Activates p53 and enhances apoptotic signaling, restores telomozide sensitivity. Downregulates p62 SQSTM and promotes autophagic flux | [46,47] |
| 10 | BC | Quinacrine | Activation of p53 signaling, suppresses NF-κB activity and apoptosis induction | [36] |
| 11 | BC | Quinacrine | Downregulation and suppression of p-CHK1/2 | [43] |
| 12 | BC | Quinacrine and SB218078(Chk1 inhibitor) | Downregulates p-Chk1/2, Cyclin $B_1$, $E_1$ and Cdc25A, arrests cells at $G_2/M$ stage, suppress BER, enhances SB218078 effect and apoptosis | [48] |
| 13 | AML | Quinacrine | Inhibits protein synthesis and ribosomal biogenesis, cell death | [49] |
| 14 | HCC | Quinacrine and TRAIL | Upregulates DR5 expression and eradicates anti-apoptotic protein Mcl-1, sensitizes cells to TRAIL mediated cell death | [50] |
| 15 | OC | Quinacrine and Rucaparib | Inhibits ribosomal biogenesis through attenuation of nucleostemin (NS/GNL3) and RPA194 expression, promotes DNA damage sensitizes cells to PARP inhibitor Rucaparib | [51] |
| 16 | OC | Quinacrine and carboplatin | Downregulates p62 SQSTM, enhances autophagic flux and restores sensitivity to carboplatin | [52] |
| 17 | OC | Quinacrine | Downregulates p62-Skp2 axis and simultaneously upregulates p21/p27 leading to autophagic cell death | [53] |
| 18 | OC | Quinacrine and TRAIL | Upregulates DR5 expression and prolongation of its half-life, sensitizes cells to TRAIL and enhances its apoptotic effect | [54] |
| 19 | OC | Quinacrine | Activates Cathespin L and downregulates p62, promotes lysosomal membrane premeability | [55] |
| 20 | RCC | Quinacrine | Activation of p53 signaling and apoptosis induction | [56] |
| 21 | HNSCC | Quinacrine and cisplatin | Restores deficient p53 expression, restores sensitivity to cisplatin and enhances its cytotoxic effect | [57] |
| 22 | HNSCC | Quinacrine and cisplatin | | [58] |
| 23 | Novikoff's Hepatoma | Quinacrine | Inhibits DNA polymerase activity of malignant cells | [59] |
| 24 | GIST | SAHA and Quinacrine | Enhances | [60] |
| 25 | GIST | | | |
| 26 | ALL | Vorinostat and Quinacrine | Multifold enhanced ROS generation and mitochondrial damage | [61] |

### 4.1. Quinacrine and DNA Intercalation

The discovery of QC's DNA binding properties and the multi-disease affectivity of this particular drug drew the attention of cancer researchers in the 1970s, which led to an exploration of QC's anti-cancer potential DNA-binding cytotoxic drugs as the standard therapeutics used for cancer treatment during that time. QC exhibits significant DNA intercalation properties similar to all other acridine backbone-based drugs, with a planar structure that allows them to intercalate through stacking between the bases. However,

intercalation is not the only way QC interacts with DNA; it also interacts with the minor groove of DNA through its side diaminobutyl chain, further stabilizing the track. DNA intercalation is thought to be the primary mode of QC anti-cancer activity. Few studies have shown nuclear fragmentation as a direct result of QC exposure in breast, gastric, and lung cancer cells, suggesting that QC may induce single-strand DNA damage [36–38]. However, this property has been proven insufficient for inducing cell death, as QC has been shown to interact with DNA through a nine-amino acid framework, but lacks an alkyl substitute to confer the additional damage necessary for triggering cell death. [62,63]. The intercalation of QC with DNA has also been reported to interfere with the methylation activity of the DNA methyl transferase enzyme DNMT1, preventing its binding to the promoter regions of multiple genes, including CDH13, E-cadherin, p16, secreted frizzle-related proteins (SRFPs), and subsequent silencing of gene expression through methylation [64].

### 4.2. Quinacrine Mediated Induction of P53 Signaling and Inhibition of NF-κB

Multiple studies investigating the molecular mechanism of cell death induced by QC have pointed out the activation of p53 signaling and simultaneous inhibition of NF-κB, independent of its DNA-damaging ability [56,63]. QC has also been shown to induce p53 stabilization in a manner different from DNA-damage-independent p53 stabilization [56,65]. QC has also been reported to restore p53 deficiency in wild-type head and neck squamous cell carcinoma cells that under express TP53 compared to normal human keratinocytes [57].

Another mechanism by which QC activates p53 is the suppression of NF-κB. The QC-mediated inhibition of NF-κB is a multicentric and multistep process. For one, its mode of action has been experimentally shown as mediating through Phospholipase A$_2$. PL A$_2$ plays an important role in the generation of platelet activation factors and eicosanoids, which are believed to play a role in NF-κB activation [66–68]. Studies have provided evidence of a link between NF-κB activation and arachidonic acid metabolites [69]. QC has been reported to block lysophosphatidic acid (LPA)-induced activation of NF-κB and transcription of its downstream targets ICAM-1 and IL-8 [70]. Another mechanism through which QC suppresses NF-κB activity is the prevention of phosphorylation of IκB kinase, which is responsible for the activation of NF-κB and its transcriptional activity [41]. Furthermore, the detailed underlying molecular mechanisms demonstrated that QC exposure confines the p65 subunit of NF-κB to the nucleus [64]. Furthermore, studies have shown that QC prevents the binding of the p65 subunit of NF-κB to the promoter region of its downstream targets, such as ICAM-1, which is necessary for the activation of NF-κB signaling [39]. QC has also been reported to primarily inhibit NF-κB activity by downregulating its phosphorylation at position 536 via IKkα. This was further confirmed by partial inhibition of H$_2$O$_2$ induced phosphorylation of the p65 subunit by quinacrine and depletion of the levels of the active subunit of facilitates chromatin transcription (FACT), which binds to the minor groove of DNA and to bent and cruciform DNA structures through its HMG domain. QC treatment has been shown to alter the DNA helix and create multiple binding sites for FACT, trapping FACT onto the chromatin, thus indicating phosphorylation of p53 and preventing the activation of NF-κB signaling due to unavailability of free active FACT [39,40,71,72]. These studies cumulatively suggest that the mechanism of QC-induced inhibition of NF-κB occurs at multiple levels and is more effective than selective inhibitors of NF-κB, which primarily inhibit activity through sequestration of the protein into the cytoplasm (Figure 1).

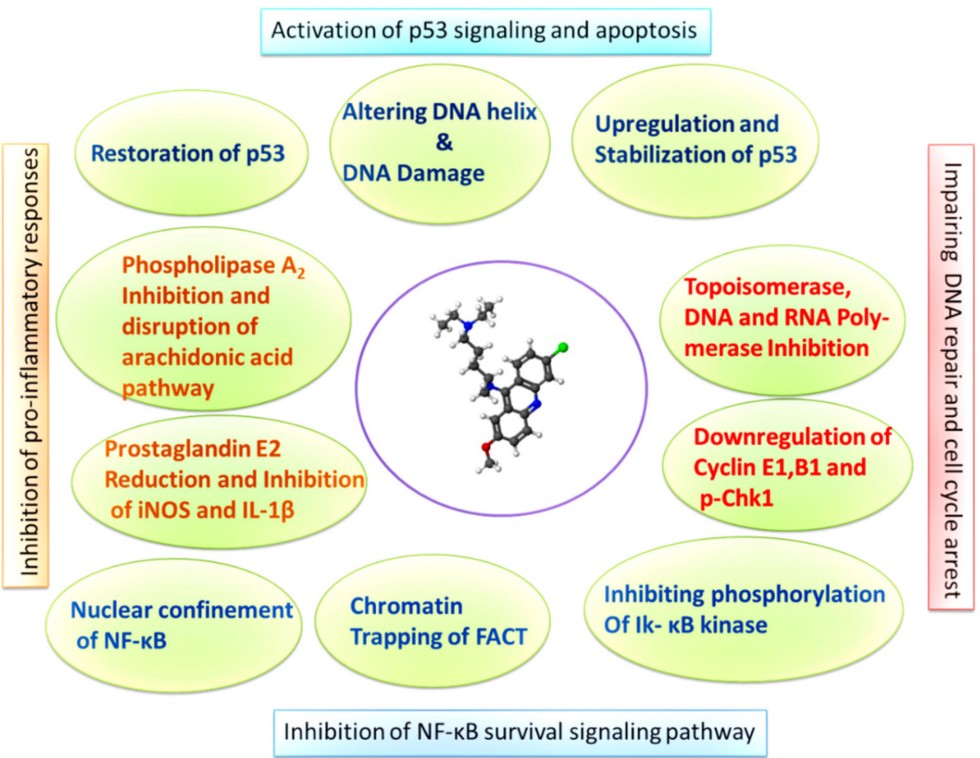

**Figure 1.** Graphic describing the molecular mechanisms of NF-κB inhibition, p53 induction, and cell cycle arrest induced by QC in cancer cells.

### 4.3. Quinacrine and Inhibition of DNA Replication Enzymes

DNA replication is central to proliferation and plays a vital role in oncogenesis. Numerous studies have provided evidence for the promiscuous nature of QC, owing to its ability to inhibit the activities of multiple enzymes that are part of DNA replication machinery such as topoisomerases, DNA polymerases, RNA polymerases, and telomerases. Numerous studies have reported the inhibition of DNA damage repair, including single-strand breaks (SSB) and base excision repair (BER) by QC, which is similar to topoisomerase inhibitors, as topoisomerases are predominantly involved in this process [73–77]. Most of these studies have demonstrated significant DNA damage repair mechanism inhibition, enhancing the toxic potential of radiation, such as UV and X-rays, or chemical inhibitors of topoisomerase II, such as etoposide. Further, the direct inhibition of topoisomerase activity occurred on breast cancer cells which showed supercoiling of plasmid DNA mixed with breast cancer cell lysate when treated with 10 and 15 μM concentrations of QC, similar to topoisomerase inhibitor etoposide, used as a positive control.

Telomerases are a family of enzymes responsible for replicating telomeres and have been associated with aggressive tumor growth and malignant growth in various types of cancer. Deregulation of its expression has been observed in many cancer types, enabling them to override replicative cell death and achieve replicative immortality. QC has been reported to inhibit purified telomerase activity in humans, *T. thermophila*, and *E. adiculetaus*, at 50 μM concentration, which is much higher than the $LD_{50}$ observed in various cancer cells [78]. However, the exact mechanism underlying this phenomenon remains unclear. Furthermore, another possible mechanism through which QC could inhibit telomere replication has been shown through stabilization of G-quadruplex structures in the telomere region, which are well known to inhibit telomeric replication in in vitro and in vivo studies [79,80].

Another aspect of QC's action on controlling DNA replication through the inhibition of DNA polymerase activity has been reported in a few studies. These studies further showed almost complete inhibition of DNA synthesis at a concentration of 32 μM QC [81,82].

Although these reported $IC_{50}$ values are much higher than the $LD_{50}$ value of the drug on cancer cells, they found their relevance in the context of cancer in experiments reported in another independent study, demonstrating a significant preference for QC in inhibiting DNA polymerases from malignant cells compared to those from normal non-transformed cells. They demonstrated that the $IC_{50}$ concentrations for DNA polymerases $\alpha$, $\delta$, and $\varepsilon$ from Novikoff's hepatoma were 15.2, 22.6, and 11.4, respectively, whereas the same $IC_{50}$ concentrations of 92.5, 200, and 146 μM for DNA polymerases $\alpha$, $\delta$, and $\varepsilon$ were isolated from normal rat liver [59]. The exact reasons for this selective preference are yet to be understood and could be one of the reasons why cancer cells are multifold more sensitive to QC than normal non-malignant cells. Furthermore, a few studies have also presented evidence suggesting that quinacrine inhibits ribosome biogenesis (RBG) by suppressing the activity of the RPA194 catalytic subunit of RNA polymerase-I and nucleolar RBG nucleostemin (NS/GNL3), inducing nucleolar stress, which leads to a reduction in RAD51 recruitment to the DNA damage site and eventually causes disruption of the homologous recombination repair mechanism [49,51].

### 4.4. Quinacrine Induced Autophagy and Cell Cycle Arrest

In addition to inducing apoptotic cell death, QC has also been reported to trigger autophagic signaling and disrupt the cell cycle, independent of DNA damage and repair activity. Studies have linked the autophagy-promoting activity of QC to its p53 upregulation effect and through downregulation of the S-phase kinase-associated protein 2 (Skp2)–p62/SQSTM axis, which is directly involved in the degradation of the tumor suppressor proteins p21 and p27. The downregulation of these autophagosome cargos triggers the accumulation of LC3B-II chains in autophagic vacuoles and initiates p21/p27 mediated apoptotic signaling [42,52,53]. Aberrant regulation of the cell cycle is a prominent promoter of uncontrolled proliferation, which is exploited by cancer cells for rapid growth. Several studies have pointed out the effect of QC on the progression of the cell cycle in breast and gastric cancer cells by altering the DNA structure, genome fragmentation, and arresting cells at the S-phase stage, presumably through multiple mechanisms, including the inhibition of DNA and RNA polymerases and topoisomerases. A few studies have also highlighted the direct downregulation of cyclin $D_1$, E, and $B_1$, and DNA polymerase subunit PCNA, thereby imparting cell cycle machinery. In addition, QC has been reported to promote the degradation of phosphorylated checkpoint kinase 1 (p-Chk1) and downregulate replication protein A (RPA), impairing base excision repair (BER) machinery. Another key event initiated by QC treatment is the phosphorylation of eIF2$\alpha$ at the Ser51 position, causing the shutdown of global protein synthesis. All these mechanisms cumulatively prevent the cell from proceeding through the stages of division [38,43,48,83,84] (Table 2).

**Table 2.** Table demonstrating the reported molecular targets of QC and its downstream effect on cellular cascades. Molecular targets of Quinacrine (QC) impacting cancer progression.

| Reported Molecular Target | QC Effect | Downstream Effect on Functionality/Pathway of Target | Reference |
|---|---|---|---|
| Gluthathione-S Transferase A1 | Inhibitor | Inhibits glutathione conjugation with electrophilic compounds and subsequent dissolution of oxidative stress caused by free radicals and alkylating drugs | [38] |
| P53 | Activation, stabilization, and upregulation | Activates the p53 signaling and stabilizes p53 leading to apoptosis | [56,64,65] |
| NF-κB | Downregulation | Inhibits NF-κB transcription activity and activation of downstream signaling, preventing initiation of cell survival mechanisms | [64,69,70] |
| Phospholipase $A_2$ | Inhibitor | Interferes with release of arachidonic acid and prevents precursor synthesis for eicosanoids | [57,66,67] |

**Table 2.** *Cont.*

| Reported Molecular Target | QC Effect | Downstream Effect on Functionality/Pathway of Target | Reference |
|---|---|---|---|
| FACT | Downregulation | Trapping FACT onto chromatin and preventing activation of downstream survival signaling molecules such as NF-κB | [39,71,72] |
| Topoisomerases I and II | Inhibitor | Impairs the helical binding activity of topoisomerases and interferes with DNA replication and repair | [40,73–76] |
| Telomerase | Inhibitor | Inhibits telomerase replication by preventing the formation of G-quadruplex structures in telomeres | [77–79] |
| DNA Polymerases α,δ, and ε | Inhibitor | Selectively inhibits the DNA Polymerase activity in malignant cells | [80–82] |
| RNA Polymerase II | Inhibitor | Interferes with ribosomal biogenesis and prevents transcription | [49,59] |
| P62/SQSTM | Downregulation | Downregulates the autophagosome cargo protein and tiggers accumulation of LC3B-II in autophagic vesicles | [42,51] |
| P21/p27 | Upregulation | Induces upregulation of p21/p27 and prevents its degradation by Skp-1 | [52] |
| CHK | Downregulation | Depleting the levels of phosphorylated Chk-1 and RPA and interefering with BER | [43,48,53] |

### 4.5. Quinacrine and TRAIL Sensitivity

TRAIL and its receptors TRAIL1/2 (also called death receptors DR4 and DR5) play vital roles in apoptotic signaling. Loss of cellular sensitivity to TRAIL, either through mutations in its downstream molecule c-FLIP or mutations in the TRAIL receptor death domain, has been shown to be one of the mechanisms through which cancer cells possibly evade cell death, and numerous studies have reported QC restoring TRAIL sensitivity in ovarian cancer, breast cancer, hepatocellular carcinoma, and glioblastoma [45,50,54,85]. QC has been demonstrated to significantly upregulate DR5 protein levels and prevent its degradation through lipid raft sequestration and autophagy. A combination of QC and TRAIL ligand at 10–40 ng/mL has been reported to induce death in over two-thirds of cells that were previously resistant to even very high concentrations of TRAIL (256 ng/mL). Another interesting mechanism through which QC facilitates TRAIL-induced apoptotic signaling is the formation of a functional bridge between TRAIL and DR5 receptors, as reported by Das et al. [86]. In addition, QC induced upregulation of p53 and p21, eradication of pro-survival Bcl family protein Mcl-1, and generation of high amounts of ROS and nitric oxide (NO) cumulatively enhanced the apoptotic effect multifold when used in combination with TRAIL.

### 4.6. Quinacrine and Chemoresistance

Chemoresistance is a major obstacle to cancer treatment. Cancer cells have been shown to evade the effects of therapeutics through various means, including effective pumping of the drug from the cell through ATP-dependent efflux pumps, such as P-glycoproteins [87], and activating alternative signaling pathways, including FGFR1, PI3K/mTOR, and Wnt-TCF, to substitute for targeted signaling blockade by therapeutics [88,89]. One of the key aspects of QC activity that has emerged over the years of extensive research is the restoration of chemosensitivity of chemotherapeutic drugs to which cancer cells have acquired resistance. Numerous studies have reported this game-changing phenomenon in which QC has restored and sensitized cancer cells to various therapeutics currently being used for the treatment of cancer. QC has been reported to reverse cisplatin chemoresistance

in head and neck squamous cell carcinoma by restoring p53 deficiency [57]. One report also demonstrated that QC restored vincristine sensitivity by preventing the efflux (outward transport) of drugs by cancer cells when exposed to vincristine [90]. Another study pointed out the restoration of sensitivity by QC through simultaneous upregulation of cathepsin L and downregulation of p62 sequestosome apparatus [55]. A few more studies have also reported the reversal of resistance to other chemotherapeutic drugs, including cytarabine in ALL xenografts, MAPK pathway inhibitors in NRAS and BRAF mutant melanoma, TRAIL in glioblastoma and hepatocellular carcinoma cells mostly through simultaneous downregulation of p62 and NF-κB, upregulation of p53, mitochondrial damage, and subsequent activation of caspases [54,85,91,92]. In addition to its aforementioned effects, QC has also been reported as an inhibitor of P-glycoproteins, thereby impairing its own efflux, as well as the combinational therapeutic administered with it [93,94]. In addition, QC has been reported to downregulate canonical Wnt signaling by depleting β-catenin and phosphorylated GSK3β levels [95]. QC has also been reported to inhibit Akt activity by preventing its phosphorylation at Ser437 and interrupting the positive feedback loop between Akt and mTOR, thereby inhibiting the Akt-mediated proliferative signaling cascade [96]. Additionally, QC has been shown to downregulate and promote JNK1 mediated degradation of nuclear factor E2-related factor 2 (Nrf2), which is known to upregulate the expression of oxidative stress-responsive genes, prevent the increase in double-stranded DNA breaks as a consequence of oxidative and electrophilic stress, and restore sensitivity to chemotherapeutics such as 5-FluoroUracil and cisplatin in colorectal and lung cancer cells [97,98]. Furthermore, recent findings from our group regarding its direct role in binding to the G-site of glutathione-S transferase subtypeA1 (GSTA1) and preventing conjugation of reduced glutathione with electrophilic compounds have added another interesting aspect of its potential for the reversal of chemoresistance [38] (Figure 2).

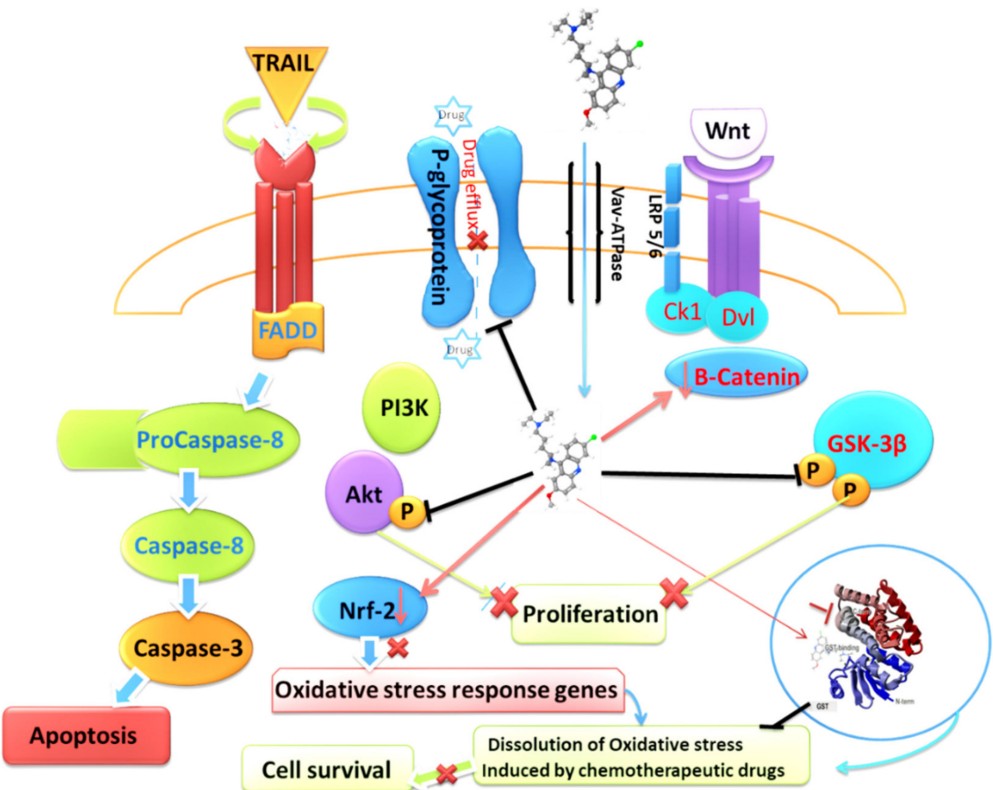

**Figure 2.** Graphic describing the effect of QC on chemoresistance pathways enacted and utilized by cancer cells.

### 4.7. Synergistic Effects of Quinacrine with Other Chemotherapeutic and Targeted Therapies

QC has been studied in combination with various clinically approved therapeutics to explore the possibility of combinational therapeutic strategies. Every study conducted so far has demonstrated QC's good synergy with cytotoxic, microtubule-targeting, and targeted therapeutic TKIs, which are unique to all chemical therapeutics. QC has shown its potential to enhance the therapeutic effect by many times and achieve the desired anticancer effect at much lower concentrations than the specific drug alone in both preclinical and clinical studies. In laboratory models, QC has shown good synergy with vorinostat (SAHA, an HDAC inhibitor), paclitaxel, 5-Fluorouracil, daunorubicin, cisplatin, cytarabine, erlotinib, and sorafenib in various types of cancer cells [44,58,60,61,99–103] (Table 3). QC has also been reported to synergize with the cytotoxic drugs carboplatin in ovarian cancer [52], temozolomide, SGK kinase inhibitor drug SI113 Glioblastoma multiforme (GBM) [46,104], and the PARP inhibitor drug Rucaparib in ovarian cancer cells [51]. QC has also been reported to enhance the effects of imatinib in GIST tumors through autophagy inhibition and to reverse the resistance to imatinib in both IV vitro and xenograft models [47]. These studies have demonstrated the remarkable effect of quinacrine in enhancing the potential of other drugs. They have opened new frontiers for exploration of even more permutations and combinations of QC with promising new targeted therapeutics, including therapeutic mAbs and immunotherapies.

**Table 3.** Table describing the synergistic potential of QC with numerous clinically approved therapies. Synergy of QC with clinically approved therapies.

| Sl No. | Drug | Nature of Combination with QC | Mode of Study | Type of Cancer | Reference |
|--------|------|------|------|------|------|
| 1 | Sorafenib | Synergistic | In Vitro and In Vivo models | Anaplastic thyroid cancer | [58] |
| 2 | Vorinostat (SAHA) | Synergistic | In Vitro and In Vivo models | Gastrointestinal cancer, T-cell acute lymphoblastic leukemia | [94,95] |
| 3 | 5- FluoroUracil | Synergistic | In Vitro and In Vivo models | Colorectal cancer | [60] |
| 4 | Paclitaxel | Synergistic | In Vitro and In Vivo models | Prostate cancer | [97,99] |
| 5 | Daunorobucin | Synergistic | | Breast cancer | [98] |
| 6 | Cisplatin | Synergistic | In Vitro | Head and neck squamous cell carcinoma, Endometrial cancer | [93,96,99] |
| 7 | Erlotinib | Synergistic | In Vitro and In Vivo models, and Phase I Clinical study | Non-small cell lung cancer | [72,105] |
| 8 | Capecitabine | Synergistic | Phase I Clinical stuy | Colorectal cancer | [106] |
| 9 | Cytarabine | Synergistic | In Vitro and In Vivo models | Acute Myeloid Leukemia (ALL) | [61,91] |
| 10 | Vincristine | Synergistic | In Vitro and In vivo models | Leukemia | [90] |
| 11 | Carboplatin | Synergistic | In Vitro and In vivo models | Ovarian cancer | [52] |
| 12 | Temozolomide | Synergistic | In Vitro | Glioblastoma Multiforme (GBM) | [104] |
| 13 | SI113 (SGK inhibitor) | Synergistic | In Vitro | Glioblastoma Multiforme (GBM) | [46] |
| 14 | Rucaparib | Synergistic | In Vitro | Ovarian cancer | [51] |
| 15 | Imatinib | Synergistic | In Vitro and In vivo models | Gastrointestinal Tumor (GIST) | [47] |

## 5. Quinacrine Nanoparticles in Cancer Treatment

Nanoparticle formulations of clinical therapeutics possess certain advantages, such as enhanced permeabilization due to their smaller size, thereby reducing the concentrations of effective dosage by multifold. Several studies have explored the aspect of assessing the formulation of QC nanoparticles (NPs), both standalone and in combination with other therapeutics, for their impact on various cancer cells. Hybrid gold and silver-based nanoparticles of QC have been reported to induce cell viability reduction, cell cycle arrest, and apoptosis at very low concentrations of 0.5 μg/mL in SCC-09 cells [107]. Another study

from the same research group showed the effectiveness of silver and gold-based hybrid nanoparticles in downregulating cancer stem cell-specific markers by 40–60%. Loss of DNA repair activity due to downregulation of the BER mechanism was also observed due to QC NPs [108]. Quinacrine-loaded *Undaria pinnatifida* fucoidan nanoparticles have been shown to enhance the effect of QC by nearly 5.7-fold and induced 68% tumor reduction in xenograft models compared to 20% with liquid drug alone in pancreatic cancer cells [109]. Another study also reported disruption of hedgehog signaling by QC NPs and induction of apoptosis at lower concentrations than the liquid formulation of the drug. They have provided evidence for QC NP-mediated inhibition of the hedgehog-GLI cascade through binding of QC NPs to the consensus sequences (5 GACCACCCA3) of GLI1 and destabilizing the cascade [110]. Nano formulations of QC and erlotinib have also been recently reported to exhibit much better synergy than their plain drug solutions in NSCLC three-dimensional models. Multifold-enhanced antiproliferative and cytotoxic effects were observed in the nano form of both drugs [111]. Another study explored the development of inhalable bovine serum albumin-coated QC NPs for the treatment of NSCLC. The nanoparticles showed good aerosolization potential, with less than 5 μm median diameter. These aerosolized NPs were found to be much more effective on different NSCLC cells in three-dimensional culture models. They observed lower $Ic_{50}$ values and prominent cell cycle arrest, leading to apoptotic death due to aerosolized NPs compared to the standard drug exposed in liquid form [112]. All these studies have laid a solid foundation for further exploration of QC NPs, both as a standalone and in combination with other therapeutics in xenografts as well as early phase clinical studies.

## 6. Clinical Research Studies of Quinacrine in Cancer Treatment

The earliest hints about the possibility of utilizing QC as a potential clinical anti-neoplastic drug came from a series of clinical studies exploring QC for the treatment of malignant pleural effusions as multiple cancers, including breast, lung, ovarian, and lymphoma, develop malignant effusions during the course of their disease. All these reports suggested a significant response of QC, leading to noticeable improvement in OS of the patients and a remarkable regression in reaccumulating of pleural fluid for months following QC treatment [14,38,113]. These studies reported that 68–72% of patients treated with a single dose or multiple oral dosages showed significant responses, and the average duration of response was approximately 20 weeks in patients without lung metastases. Taylor et al. also reported longer survival (<1 year) in 28% of the patients without recurrent effusion. Later, another study exploring QC combination and tetracycline in a randomized trial was conducted by Bayly et al. [114]. Six out of ten patients who received QC demonstrated a complete response to the therapy, with OS survival ranging from 3 to 6 months, and three other patients showed a partial response to the treatment.

Later, a few studies presented evidence of a decrease in fibrinolytic activity, evident by decreased levels of D-dimer, and fibrin lysis was observed after six hours of QC treatment. The levels of regulators of fibrinolysis, plasminogen activator inhibitor PAI-1, thrombin, fibrinopeptide A(FPA), and beta-thromoglobulin markedly increased after QC treatment. High levels of these markers, especially FPA and thrombin, are indicative of active fibrin formation and play an important role in arresting the pleural exudation process [115,116]. A single-agent quinacrine phase II clinical trial for androgen-dependent prostate cancer was conducted at the Cleveland Clinic, USA (clinical trial identifier NCT00417274). The patients were administered 100 mg of the drug daily and monitored for 18 months. Only three out of 31 patients showed serious adverse side effects (one back pain, one sepsis, and one thrombosis). The other details of this study are yet to be elucidated. Two other phase I clinical studies explored QC's efficacy as a combinational therapeutic approach. A phase I dose-escalation clinical trial of a QC and erlotinib combination in advanced stage metastatic NSCLC (clinical trial identifier NCT0183995) recommended a dose of 50 mg QC every other day with 150 mg erlotinib for further phase II studies, which are currently in the planning stage. The dose was well tolerated with mild and manageable side effects. One

of the six patients showed stable disease (SD) for eight months, and one patient showed a partial response (PR) for six months duration [105]. However, all of the patients selected for this study either had EGFR-negative adenocarcinoma or squamous cell carcinoma, which technically could limit the effects of erlotinib, as shown in various earlier phase II/III studies of erlotinib in EGFR negative and SCC patients. Another phase Ib clinical study completed this year assessed the combination of quinacrine and capecitabine in metastatic colon cancer patients (clinical trial identifier NCT01844076) [106]. Three patients showed SD and one showed PR in the dose-finding cohort of the study, while two out of the seven patients in the expansion cohort showed SD on 100 mg/kg QC. The ORR for the entire cohort was 5.9%, the disease control rate (CR + PR + SD) was 35.4%, and the OS in the heavily pretreated overall cohort was approximately 5.22 months. Interestingly, four of the six patients with responsive disease exhibited mutations in the K-Ras gene. The study concluded that a safe and well-tolerated dose of 100 mg QC and 1000 mg/m$^2$ capecitabine was required. Although both Phase I studies have shown promising results, no Phase II studies have been reported yet.

### 7. Conclusions

These recent preclinical and clinical studies have put QC into the spotlight for exploring combinational therapeutic regimens that are more effective, less prone to chemoresistance, and have overall tolerable toxicity profiles. Additionally, a new generation of drugs, such as CBL 0137, that are modelled on QC have been designed and found effective in pre-clinical studies and currently have entered phase I clinical studies (NCT04870944). More clinical studies continuing this trend in NSCLC and other cancers are warranted. Moreover, the polypharmacological potential of quinacrine, which is able to simultaneously target multiple signaling cascades and cellular processes that are implicated in promoting carcinogenesis, is indeed a beneficial factor, adding a unique aspect to its scope of utility for cancer treatment. Simultaneously, more research is required to further elucidate the role of QC in cancer regression, and the molecular mechanisms involved and their synergy with new innovative targeted therapies are required to fully utilize the polypharmacological potential of this small molecule in combatting cancer. Numerous studies on nanoformulations of QC have opened a new frontier for further increasing the bioavailability of the drug and have further reduced the concentrations required to achieve remarkable antineoplastic effects. Further intensive exploration of new formulations and combinations of QC NPs with nanoformulations of other clinical therapeutics will provide less toxicity and better new promising therapeutic formulations and strategies in the future.

**Author Contributions:** Conceptualization, A.S. and M.K.; methodology, M.K.; formal analysis, M.K.; data curation, M.K.; writing—original draft preparation, M.K.; writing—review and editing, A.S.; supervision, A.S.; project administration, A.S.; funding acquisition, A.S. All authors have read and agreed to the published version of the manuscript.

**Funding:** No specific funding was received for this study.

**Institutional Review Board Statement:** Not applicable.

**Informed Consent Statement:** Not applicable.

**Acknowledgments:** Authors thank Goa Cancer Society, Goa, India and DBT, Govt. of India for financial support. M.K. acknowledges the UGC and the Govt. of India for his fellowships.

**Conflicts of Interest:** The authors declare no conflict of interest.

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
