# Peer review of "Repurposing of Anti-Malarial Drug Quinacrine for Cancer Treatment: A Review"

_scipharm, doi:10.3390/scipharm90010012_

Round 1

Reviewer 1 Report

The authors comprehensively review the repurposing potential of quinacrine in cancer treatment. While other groups have focused on quinacrine potential against cancer stem cells and chemotherapy resistance, this seems to be the first comprehensive review of quinacrine experimental use for oncology in over a decade. The manuscript provides a good historical background for quinacrine and gives credit to a very early study of quinacrine in cancer. The methodology for reviewing the relevant literature is clearly stated. The main body of the manuscript categorizes quinacrine pharmacology studies into various oncology pathways and specific research areas.

Overall, this is an important review that was generally well executed, but there are a few concerns. This major concern is improper citations/numbering, where it is likely that the numerical order of the bibliography has errors. Furthermore, expanding the review by discussing studies in a few key areas will improve the comprehensive nature of this manuscript, and some minor corrections are noted.

  1. The references in the bibliography are appropriate from what can be assessed, but citation numbers in the text do not match up to a relevant bibliography reference starting with reference 3 (#3-6 are not full references). Please carefully review citations and reorder/fix/update any errors between citation numbers and the bibliography order.
  2. [lines 113-119, and section 5] Quinacrine was evaluated for the management of pleural effusions, but cytotoxicity to cancer cells was not found (or assessed in many cases) across many studies. To call these “explorations of anti-cancer potential” is misleading. Revise the descriptions and conclusions reported here, and specifically review and cite evidence of cytotoxicity to cancer cells separately (if there are studies to support this).
  3. The authors and past articles note the polypharmacology aspects of quinacrine. It would be helpful to discuss this phenomenon since it relates to the manuscript sections and add a statement in the conclusion regarding this. Are the actions of quinacrine context/cell dependent? Is the polypharmacology due to DNA binding in various genes and/or epigenetic effects (e.g. as a epigenetic methyltransferase inhibitor PMID: 23593653)?
  4. The section on chemotherapy resistance is limited and should be expanded with more studies relevant to quinacrine in chemotherapy resistance models. This includes, but is not limited to, studies that the authors cited in other sections of the manuscript that applied quinacrine to drug resistant/refractory cell lines and xenografts.
  5. The synergistic effects of quinacrine with other drugs seems like it is missing a some studies. Specifically, there are papers on synergistic combinations in more breast cancer studies, glioma, mesothelioma, and ovarian cancer with various other drugs that could be mentioned. The corresponding table could include more combinations with approved drugs such as carboplatin, pemetrexed, and olaparib.
  6. Other minor comments by line(s):

[lines 10-11] “Since its discovery in the early 1950s…” implies that quinacrine was discovered in the 1950s, which is not correct.

[line 34] Statement on Plasmodium falciparum is not clear.

[line 145-147] Reference the “multiple studies”

[line 222] RAD5 should be RAD51

[line 327-329] References needed for these “few reports”

[line 341] If phase II studies are being planned, a recent reference or the source of information should be included. Otherwise it would be more accurate to state that any start of Phase II trials has not been reported.

Reviewer 2 Report

Reviewer’s comment:

The review article “Repurposing of anti-malarial drug quinacrine for cancer treatment: A review” by Makhan Kumar and Angshuman Sarkar encompasses almost all scientific findings and relevant articles in PubMed. 

This review article is broadly categorized into 

- Pharmacokinetics and toxicities of QC

- Molecular targets of QC impacting cancer progression

- QC mediated alteration in chemoresistance 

- QC facilitated synergistic effects of known chemotherapeutics

- Clinical studies of QC

This systemic study is almost thoroughly documented and subsequently indexed. However, some critical points have yet to be addressed for further consideration in this journal.

Comments:

  1. The author should consider making a table based on the types of cancers where the effect of QC has been studied.
  2. There is a recent review from Dr. Shridhar’s group [Semin Cancer Biol. 2021 Jan;68:21-30]. The authors should address how their review article is procuring better insight on repurposing QC than the above-mentioned article? The authors should also cite that article.
  3. In “Molecular targets of QC impacting cancer progression” section authors should also incorporate another recent observation QC mediated Cathepsin L activation and subsequent cell death [Cancers (Basel) 2021 Apr 21;13(9):2004].
  4. The authors should incorporate another broadly studied category “QC and nanoparticle”
  5. Please be consistent with the Clinical trial identifier number and mention it in section 5 as already mentioned in line no. 334-335

Minor comments:

  1. Please be consistent with the abbreviation. If you are writing ‘quinacrine’ in subheading then be stable with that (e.g. subheading 4.5).
  2. Please revise the whole manuscript for grammatical errors.  

Round 2

Reviewer 1 Report

Changes are sufficient and review article is a good contribution to the field.

Reviewer 2 Report

The authors have made changes in the article almost all as suggested.